# Succeed or Learn Slowly: Sample Efficient Off-Policy Reinforcement Learning for Mobile App Control

**Georgios Papoudakis[1*]**   **Thomas Coste[1*]**   **Jianye Hao[1]**   **Jun Wang[2]**   **Kun Shao[1†]**

[1]Huawei Noah's Ark Lab, [2]University College London

## Abstract

Reinforcement learning (RL) using foundation models for policy approximations in multi-turn tasks remains challenging. We identify two main limitations related to sparse reward settings and policy gradient updates, based on which we formulate a key insight: updates from positive samples with high returns typically do not require policy regularisation, whereas updates from negative samples, reflecting undesirable behaviour, can harm model performance. This paper introduces Succeed or Learn Slowly (SoLS), a novel off-policy RL algorithm evaluated on mobile app control tasks. SoLS improves sample efficiency when fine-tuning foundation models for user interface navigation via a modified off-policy actor-critic approach, applying direct policy updates for positive samples and conservative, regularised updates for negative ones to prevent model degradation. We augment SoLS with Successful Transition Replay (STR), which prioritises learning from successful interactions, further improving sample efficiency. We evaluate SoLS on the AndroidWorld benchmark, where it significantly outperforms existing methods (at least 17% relative increase), including prompt-engineering and RL approaches, while requiring substantially fewer computational resources than GPT-4o-based methods with 5-60x faster inference.

## 1 Introduction

Mobile phones have become a central part of daily life, supporting communication, productivity, financial management, and entertainment. As mobile apps become more complex, there is a growing need for automated systems that can understand and interact with mobile interfaces. Such systems could improve accessibility for people with disabilities, enhance automated testing, and act as assistants capable of completing multi-step tasks on behalf of users.

Reinforcement learning (RL) offers a promising approach to training agents for mobile app interaction [e.g., 2], allowing learning through trial and error without requiring extensive manual labelling. However, applying RL to mobile app control presents several unique challenges. First, like many open-ended tasks, these environments often provide sparse or no reward signals. Second, the action space is large and context dependent, requiring the agent to choose to interact from hundreds of possible UI elements and many different action types. Third, simulations in mobile environments are time and computationally expensive: several seconds to execute each step limits training interactions.

Foundation models have demonstrated remarkable capabilities in understanding text instructions and visual context, making them promising candidates for mobile app control. Recent efforts have shown that prompting LLMs such as GPT-4o can yield agents capable of following instructions to complete tasks in mobile environments [14, 28, 36, 38]. However, these approaches typically require multiple

---

\* Equal contribution.

† Correspondence to kun.shao@huawei.com

39th Conference on Neural Information Processing Systems (NeurIPS 2025).

API calls per step and extensive in-context reasoning, resulting in high operational costs and slow inference times. Meanwhile, Supervised Fine-Tuning (SFT) approaches that directly train smaller models on human demonstrations struggle with generalisation to new and online tasks [9, 21, 23, 35].

In this paper, we introduce Succeed or Learn Slowly (SoLS), a novel RL algorithm that enables sample-efficient fine-tuning of foundation models. This work tackles the challenges of sparse rewards and limited samples through two complementary innovations. Our first contribution is a dynamic policy update mechanism that aggressively reinforces successful actions while conservatively regularising unsuccessful ones, helping prevent performance degradation and forgetting during exploration. This pairs with our second innovation, Successful Transition Replay (STR), which selectively stores and prioritises successful state-action pairs from previous episodes. Together, these approaches create a sample-efficient learning system that extracts maximum value from limited generated samples, substantially improving the model's ability to learn effective control strategies even with minimal training data.

We evaluate SoLS on AndroidWorld [20], a benchmark of real-world mobile app control tasks across three difficulty levels. Our approach achieves an overall success rate of 51.3%, significantly outperforming both expensive GPT-4o-based approaches and other RL-based fine-tuning methods. Importantly, our method requires only a single pass through an 8B parameter model per step, resulting in inference times approximately 5-60x faster than state-of-the-art prompting approaches.

Our contributions can be summarised as follows:

- We introduce SoLS, a novel off-policy RL algorithm specifically designed for sample-efficient fine-tuning of foundation models in sparse reward settings, which achieves the best performance, by at least 17% relative increase, and the fastest inference time of all baselines.

- We propose Successful Transition Replay (STR), a technique that prioritises learning from successful interactions to maximise learning efficiency in environments with costly simulations.

- We show that small language models, when fine-tuned with suitable RL techniques, can outperform much larger foundation models on mobile app control tasks, and offer a comparative analysis of RL methods, highlighting the strengths and weaknesses of various methods.

This work helps narrow the gap between costly but effective large model prompting approaches and efficient but traditionally less performant fine-tuned models, providing a potentially practical direction for mobile app control that balances performance, efficiency, and resource requirements.

## 2 Preliminaries

### 2.1 Problem Formulation

We formulate the Mobile App Control problem within a Partially Observable Markov Decision Process (POMDP) framework, represented as $(\mathcal{S}, \mathcal{A}, \mathcal{O}, R, P, \Omega)$. $\mathcal{S}$ denotes the state space, $\mathcal{A}$ the action space, and $\mathcal{O}$ the observation space. The reward function $R$ is binary, signalling successful episode completion. Functions $P$ and $\Omega$ represent the state and observation transition processes, respectively. Episode length is bounded by a task-specific horizon $H$. An episode ends either with the completion of the goal or when $H$ is reached without success. We define the return as the terminal reward and aim to learn a parameterised policy $\pi_\theta$ that maximises the expected return across the task distribution, with $g$ sampled from the task set $\mathcal{G}$, and $r$ the episode-terminal reward:

$$\max_\theta \mathbb{E}_{g \in \mathcal{G}, \pi_\theta} [r] \tag{1}$$

Our approach leverages an offline dataset $\mathcal{D}_{off}$ containing human demonstrations, which we use to warm-start the policy $\pi_\theta$ and familiarise it with the environment's action and observation space. Additionally, we highlight that $\mathcal{D}_{off}$ is out-of-distribution (OOD) relative to the target environment. Our training pipeline follows a two-phase procedure: (1) SFT using $\mathcal{D}_{off}$, followed by (2) RL fine-tuning to optimise the objective in Equation (1). Notation used in equations throughout can be found summarised in Table 2.

---

Name inspired by WoLF [3]

## 2.2 Mobile App Control Data

Our work focuses on two open-source phone navigation datasets and benchmarks, namely Android-Control [20] and AndroidWorld [28]. AndroidControl provides an extensive training dataset of more than 13k tasks spanning 833 Android applications, including task instructions, screenshots, and UI element trees, as well as human-selected action demonstrations. We use this dataset for initial fine-tuning, leveraging the fact that its action space has strong similarities with AndroidWorld's. AndroidWorld is a benchmark consisting of 116 tasks with a different app and goal distribution compared to AndroidControl. We specifically use an 80-task subset, omitting tasks such as Q&A and verification, due to our agents and action space, as well as evaluation process (see Appendix A.4.2). Our final benchmark thus has a harder overall difficulty distribution. AndroidWorld works by connecting agents to an Android phone emulator to run tasks online, in a realistic environment with ground-truth rewards. Agents are provided with the task goal, current screenshot, and UI tree information at every step, and are required to provide actions to be executed in the environment. We use AndroidWorld as our RL environment and evaluation benchmark throughout our experiments.

It is important to mention that AndroidWorld is out-of-distribution (OOD) compared to the Android-Control dataset used for SFT. While the task domains of both overlap, almost all AndroidWorld apps are unseen. Moreover, the phrasing of goals is different, with AndroidControl goals being quite wordy and AndroidWorld goals being more imperative. Finally, though the action spaces are similar, the distribution or use of actions in certain cases can be quite different. For example, the `long-press` action is extremely rare in AndroidControl, appearing only as 0.2% of the training set, while it is an important action featured regularly in AndroidWorld, such as to clear text in a text field.

## 3 Methodology

### 3.1 Observation and SFT Step

An important design detail for experiments with AndroidWorld tasks is the construction of the observation. In this work, we use Llama-3-8B-Instruct [12] for approximating the policy. As input to the model, we use the textual goal, and process the UI tree into a list of available UI elements with descriptions and relevant attributes. Text-only input has shown potential in achieving similar or even better results than visual input [28], and requires many fewer tokens, leading to faster inference times.

As a first step, we fine-tune our base model through SFT on AndroidControl, which shares observation and action space similarities with AndroidWorld. More details on observations, action space, and output format can be found in Section 2.2 and Appendix A.4. Using a similar input format and action space during SFT and RL training, the SFT step allows the model to adapt to action-prediction in Android environments, with the required output format. The resulting SFT model works as a starting point for the RL methods, such that some tasks can be solved in the initial phase of training.

### 3.2 Successful Transition Replay

Generating trajectories in open-world environments presents significant challenges, primarily due to the high computational cost involved. For instance, in Android emulators, each forward step in the simulation can take 4-5 seconds, which makes trajectory generation resource-intensive and inefficient. Moreover, the generated trajectories tend to be long and involve many steps, but only a small fraction are successful. As a result, much of the computational effort invested in trajectory generation yields little in terms of meaningful learning outcomes.

In addition, while RL algorithms are commonly used in large-scale post-training of LLMs, most of these algorithms are designed for single-turn tasks, such as solving mathematical problems or question-answering, where reward models provide immediate feedback. However, our setting is more complex. The agent seldom solves new tasks due to a significant distribution shift from the original SFT phase, which makes traditional RL approaches less effective in our context.

To address these challenges, we propose Successful Transition Replay (STR), an experience replay [22] that stores individual successful timesteps. Figure 1 shows how STR integrates into our pipeline. It focuses on preserving only those transitions from episodes where the agent takes successful actions, ensuring that learning incorporates proven effective steps rather than failed or non-meaningful ones. By maintaining a pool of these successful timesteps, the agent can revisit and build on past successes,

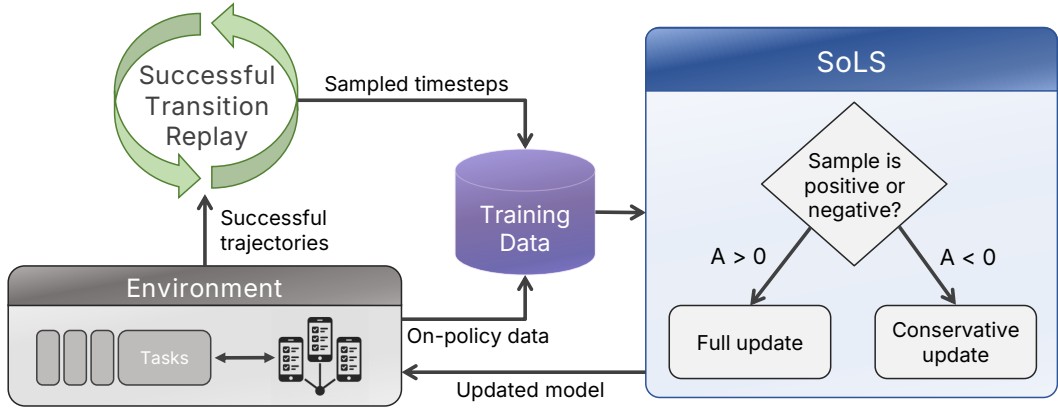

Figure 1: Illustration of the SoLS and STR methods.

ultimately leading to more efficient learning rather than relying exclusively on on-policy data. STR thus creates a bridge between the SFT and target distributions, by bootstrapping from rare successes to build a repository of successful interactions specific to the environment.

STR uses a hash table to map each task to a list of successful individual timesteps. During training, we sample a specific number of timesteps from each task and combine them with on-policy data. Each task-specific list stores the 50 most recent successful timesteps for that task. Furthermore, the training pipeline employs a data-parallel architecture, with each parallel process maintaining its own instance of STR. Let $\mathcal{D}_{\text{STR}}$ represent STR, $n$ denote the number of sampled timesteps per task, and $\mathcal{D}_{on}$ be the on-policy sampled data. The training dataset $\mathcal{D}$ for each update is generated as follows:

$$\mathcal{D} = \bigcup_{t \in \text{tasks}} \text{sample}(\mathcal{D}_{\text{STR}}(t), n) \cup \mathcal{D}_{on} \qquad (2)$$

With STR, we focus training on proven, effective actions, reducing the need for inefficient trajectory generation and improving exploration efficiency in complex, open-world environments.

### 3.3 Succeed or Learn Slowly

We highlight two critical observations about RL training for foundation models.

First, standard RL from human feedback (RLHF) techniques use policy regularisation to prevent reward model hacking. However, this concern diminishes in environments with structured action formats. Unlike free-form generation, where models might produce deceptive but incoherent text, structured environments naturally constrain actions through predefined formats. These constraints ensure outputs remain coherent and valid within the environment's action space.

Second, the actor-critic policy loss function $-A_t \cdot \log \pi(a_t \mid s_t)$ creates unintended consequences when fine-tuning models with large output vocabularies. When the advantage is negative ($A_t < 0$), the policy decreases the probabilities of tokens that formed poor actions. This increases probabilities for other tokens across the model's vocabulary; tokens that may be semantically related but inappropriate for the specific action space. This redistribution disrupts the model's learned representations. This explains the rise of rejection sampling methods in multi-turn RL literature, such as DigiRL for app control, as they avoid destructive negative updates while still improving policy performance.

We start with the off-policy actor-critic algorithm [11], assuming data is sampled from a behavioural policy $\pi_b$, used to generate the training data. We write the objective with the state distribution $d^{\pi_b}(s)$ under the policy $\pi_b$, subject to the learning policy being close to the base policy:

$$\max_{\theta} \sum_s d^{\pi_b}(s) V^{\pi_\theta}(s) \quad \text{subject to} \quad \mathbb{I}\left[A^{\pi_\theta}(s) < 0\right] \cdot \text{KL}(\pi_\theta || \pi_b) \le \epsilon \qquad (3)$$

Following the work of Degris et al. [11],

$$\nabla \mathcal{J}(\theta) = \sum_s d^{\pi_b}(s) \nabla V^{\pi_\theta}(s) \approx \sum_s d^{\pi_b}(s) \sum_a \frac{\nabla \pi_\theta(a|s)}{\pi_b(a|s)} Q^{\pi_\theta}(s,a) \qquad (4)$$

We also subtract the state value from the state-action value to reduce variance. The state value is computed under the behavioural policy $\pi_b$, which does not change with actions and therefore does not introduce bias into the policy gradient.

$$\nabla \mathcal{J}(\theta) = \sum_s d^{\pi_b}(s) \sum_a \frac{\nabla \pi_\theta(a|s)}{\pi_b(a|s)} (Q^{\pi_\theta}(s,a) - V^{\pi_b}(s)) \tag{5}$$

Additionally, we replace the state-action value with the Monte Carlo return $R$ in Equation (6) to reduce the bias of the estimator, especially early on when the value function is randomly initialised. This aligns with the work of Degris et al. [11], where the state-action value is replaced with $\lambda$-returns under the behavioural policy.

To ensure consistency with the objective in Equation (1), we apply a PPO-like [29] cut-off in the policy gradient update when the advantage is negative. This restricts policy updates when the advantage is negative and the importance sampling ratio falls outside the allowed range, preventing the policy from deviating too far from the policy that generated the action. The gradient of the loss function for the actor, with $A(s,a) = R - V^{\pi_\theta}(s)$, is therefore:

$$\nabla \mathcal{L}_{ac} = \begin{cases} -\mathbb{E}_{s,a \sim \hat{D}} \left[ A \cdot \frac{\nabla \pi_\theta(a|s)}{\pi_b(a|s)} \right] & \text{if } A > 0 \text{ or } \left( 1 - \epsilon \leq \frac{\pi_\theta(a|s)}{\pi_b(a|s)} \leq 1 + \epsilon \right) \\ 0 & \text{otherwise} \end{cases} \tag{6}$$

In contrast to the original PPO objective, which restricts policy updates when $A < 0$ and only when the importance sampling ratio falls below $1 - \epsilon$, our approach introduces a symmetric constraint by also restricting updates when the importance sampling ratio exceeds $1 + \epsilon$.

The baseline critic is trained using the same off-policy data. The value function is represented by adding an affine layer followed by a sigmoid activation on top of the final hidden layer of the transformer. We use Monte Carlo targets to avoid the need for a separate target network, which would significantly reduce the computational efficiency of SoLS. The value network parameters are updated by minimising the squared TD-loss:

$$\mathcal{L}_{cr} = \mathbb{E}_{R,s \sim \mathcal{D}} \left[ (R - V_\phi^{\pi_\theta}(s))^2 \right] \tag{7}$$

We refer to this algorithm as Succeed or Learn Slowly, or SoLS for short, which aims to update the policy following the off-policy actor-critic objective when the advantage is positive, while it follows a PPO-like regularisation in the policy updates when the advantage is negative. A high-level illustration of SoLS is shown in Figure 1. SoLS can be combined with STR without pruning the updates of positive samples due to small or large values in the importance sampling. Finally, we optimise the joint loss function by adding the two losses:

$$\mathcal{L} = \mathcal{L}_{ac} + \lambda \cdot \mathcal{L}_{cr} \tag{8}$$

## 4 Experiments

### 4.1 Evaluation Baselines

#### 4.1.1 Prompting and fine-tuned methods

**GPT-4o pure prompting methods:** First, we compare SoLS with three baseline agents that exclusively leverage large foundation models, such as GPT-4o. T3A and M3A [28] both employ a two-step prompting process: the agent summarises the previous observation, and then proposes an action based on this summary and the current observation. The primary distinction between T3A and M3A lies in the input format: T3A relies only on the UI accessibility tree, while M3A also uses a screenshot, annotated with bounding boxes around each UI element. Additionally, we evaluate SeeAct [38], which also performs two-step prompting, first generating a high-level output, then a grounded action.

**UGround [14]:** We evaluate two of the UGround-V1 models, UGround-V1 2B and 7B. UGround combines aspects of the planner-grounder SeeAct framework [38] and the actor-summariser M3A [28], along with a fine-tuned grounding model. The agent prompts GPT-4o for a high-level action, grounds the action with the UGround model, and then creates a summary by prompting GPT-4o.

**AriaUI [36]:** AriaUI follows a similar three-step planner-grounder-summariser architecture to UGround, with a fine-tuned 24.9B mixture-of-experts Aria [19] model (3.9B active parameters) as the grounder and GPT-4o as the planner and summariser.

**OS-Atlas [35]:** We use the OS-Atlas-Pro-7B variant of OS-Atlas, trained on a greater number of datasets. It has a two-stage training approach: first, GUI grounding pre-training on a custom corpus of 2.3 million screenshots, and then action fine-tuning on agent datasets such as AndroidControl. OS-Atlas-Pro-7B uses Qwen2-VL-7B [33] as a backbone, and takes as input the current goal, previous actions, and the current screenshot. A large prompt also describes the current task and action space, including custom action descriptions tailored to each dataset.

### 4.1.2 RL methods

**PPO:** Proximal Policy Optimisation (PPO) [29] is an RL algorithm that uses a heuristic clipping mechanism to prevent the training policy from deviating too much from the prior distribution. PPO can perform multiple consecutive epochs of updates using the same batch of generated data, effectively extracting more learning signal from each environment interaction while the clipping mechanism ensures the policy does not change too drastically between updates.

**A2C-STR:** Advantage Actor-Critic (A2C) [26] is an on-policy actor-critic algorithm. We augment it with STR by incorporating transitions from previously successful episodes into training updates. Since A2C is designed for on-policy learning, using off-policy data from the STR buffer requires correction. Following Degris et al. [11], we apply importance sampling to properly weight these off-policy transitions and maintain unbiased gradient estimates. Notably, A2C-STR is equivalent to SoLS when a transition has a positive advantage. However, unlike SoLS, A2C-STR continues to fully update parameters even when the advantage is negative, rather than clipping these updates.

**DigiRL-STR:** DigiRL [2] is an off-policy variant of rejection sampling [15] that evaluates transitions based on their advantage rather than their total return. We augment DigiRL with STR to leverage previously collected successful experiences, instead of the original prioritised replay buffer that was used in the original work, to ensure consistency. Further details can be found in Appendix A.3.

### 4.2 Results

Table 1 presents the success rates of SoLS and the baseline methods. For models that do not require proprietary access, we average results over three evaluation runs and report two standard errors for the overall success rate. Our results show that SoLS achieves the highest overall success rate on AndroidWorld, outperforming all baseline methods.

Table 1: Success rates of different agent methods in the AndroidWorld environment, across task difficulty levels. Overall results for non-GPT-4o methods show average success rates with two standard errors of the mean across three runs.

| | Method | Input Type | Success Rate ↑ | | | Overall Success Rate ↑ |
|---|---|---|---|---|---|---|
| | | | Easy | Medium | Hard | |
| GPT-4o | SeeAct | screen + UI tree | 36.1 | 17.9 | 0.0 | 22.5 |
| | T3A | UI tree | 66.7 | 21.4 | 12.5 | 40.0 |
| | M3A | screen + UI tree | 61.1 | 21.4 | 6.3 | 36.3 |
| FT / Mixed | GPT-4o + UGround-2B | screen | 63.9 | 25.0 | 6.3 | 38.8 |
| | GPT-4o + UGround-7B | screen | **69.4** | 28.6 | 12.5 | 43.8 |
| | GPT-4o + AriaUI | screen + UI tree | 66.7 | 28.6 | 6.3 | 41.3 |
| | OS-Atlas-Pro | screen | 40.7 | 11.9 | 6.3 | $23.8 \pm 1.2$ |
| | SFT | UI tree | 38.9 | 9.5 | 4.2 | $22.1 \pm 2.7$ |
| RL | A2C-STR | UI tree | 52.8 | 17.9 | 2.1 | $32.1 \pm 0.7$ |
| | PPO | UI tree | 53.7 | 8.3 | 6.3 | $28.3 \pm 0.7$ |
| | DigiRL-STR | UI tree | 55.6 | 32.1 | 12.5 | $38.8 \pm 0.0$ |
| | SoLS-STR | UI tree | 68.5 | **40.5** | **16.6** | **$51.3 \pm 1.2$** |

As a method simply fine-tuned on AndroidControl, SFT performs quite poorly, solving mostly easy tasks that are closely related to its training dataset, for example, those related to system apps, such as turning on/off Bluetooth, Wifi, etc. OS-Atlas-Pro achieves slightly better performance, benefitting from a GUI-grounding pre-training phase and larger range of fine-tuning datasets. Nevertheless, success rate remains limited and much lower than other methods which are able to have a better understanding of the task environment, either through larger priors or better domain learning.

In general, we observe that most methods using GPT-4o tend to perform similarly, in the 36%-44% range, with methods that use a grounding model achieving the highest performance. Methods using GPT-4o seem to perform best on easy tasks, while struggling more with medium and hard tasks compared with the strongest RL methods. This is likely because GPT-4o's strong prior and generalisation abilities allow it to understand and solve easier tasks well, while harder tasks usually require more in-domain and specific knowledge, where GPT-4o's assumptions and reasoning might not be as useful or sufficient. We also note that while UGround-2B performs worse than UGround-7B, it still performs quite well, leveraging GPT-4o's capabilities.

Among RL methods, DigiRL-STR achieves the highest performance among existing approaches with 38.8% overall success rate. However, our proposed SoLS-STR significantly outperforms all baseline methods, achieving 51.3% overall success rate. This represents a 32.5% relative improvement over DigiRL-STR and significantly exceeds even the best GPT-4o-based methods. The improvement is particularly notable on medium difficulty tasks, where SoLS-STR achieves 40.5% compared to DigiRL-STR's 32.1%. Finally, PPO achieves the lowest success rate overall among the RL algorithms, which highlights the need for STR, as is the only algorithm that is trained on on-policy data.

To further highlight the contribution of our asymmetric update mechanism, we compare SoLS with standard A2C using identical STR parameters. The results show that SoLS-STR improves upon A2C-STR by approximately 60% relative improvement (51.3% vs 32.1%). This dramatic improvement validates our core hypothesis that conservative updates for negative-advantage actions prevent performance degradation while maintaining aggressive learning from positive experiences.

## 4.3 Additional Studies

First, we evaluate the effect of STR on the success rate of different RL algorithms. For this comparison, we augment PPO with STR and, as such, enable the reuse of successful trajectories, transforming our implementation into an off-policy variant. This off-policy nature occasionally results in gradient clipping for samples retrieved from the STR buffer due to the policy divergence constraints. Figure 2a presents the success rates of PPO, DigiRL, and SoLS with and without STR. Our results show that STR significantly improves the success rate for both algorithms. We also observe that SoLS outperforms

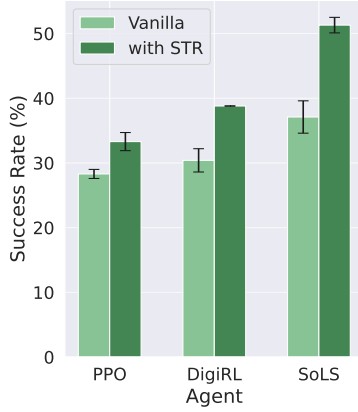

(a) Success Rate with/without STR

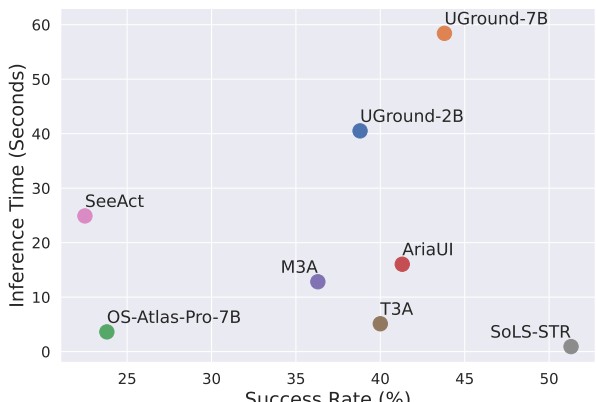

(b) Success Rate vs Inference Time

Figure 2: *Left*: Bar plot presenting the average success rate and two standard errors of the mean for PPO, DigiRL and SOLS with and without STR. *Right*: Scatter plot illustrating the trade-off between success rate and inference time. The most desirable location is in the bottom-right, demonstrating strong success rate and low inference time, which SoLS occupies.

PPO and DigiRL even when all three are trained exclusively on on-policy data, confirming our hypothesis that the asymmetric policy updates lead to higher success rate on AndroidWorld. Notably, SoLS outperforms even PPO-STR, and is on par with DigiRL-STR.

Figure 2b illustrates the trade-off between success rate and inference time across different models. SoLS occupies the best position in the bottom-right quadrant, achieving the highest success rate (51.3%) while maintaining the fastest inference time ($\sim 0.9$ seconds). This represents approximately a 60× speedup compared to UGround-7B, the next-best performing agent, which requires 40-60 seconds per step due to its multi-step prompting pipeline and slow grounding model. The efficiency gain is crucial for practical deployment in real-world scenarios where response time is critical.

In Figure 3, we show how the success rate of SoLS evolves between the first part of training and the end. For every task category, the success rate at the end of training is higher than near the beginning, showing the importance of RL training. This difference in success rate is less noticeable for certain task categories, such as Audio or System, where the early success rate is already high. This is because the base SFT model already performs quite well in these categories due to similarities with the training set, and early training steps also reinforce this behaviour, such that further training only marginally improves performance. In other categories, particularly Files, Maps, and Markor, the increase in success rate is very large, as the agent learns to solve tasks from these categories during exploration and reinforces them throughout training, especially with the help of STR.

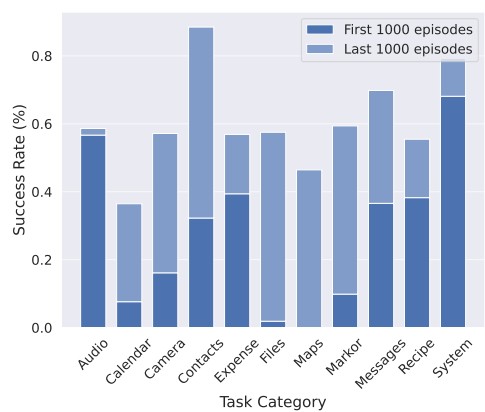

Figure 3: SoLS success rate comparison at the beginning and end of training, by task category.

## 4.4   Failure Case Analysis

Most of the tasks that SoLS consistently fails to solve can be grouped into four categories: tasks requiring memory, tasks requiring visual input, tasks requiring unseen interactions, and long-horizon tasks. We provide further explanation below, as well as some case studies in Appendix B.2.

Some tasks require the agent to remember information from previous steps, such as numbers or text displayed in earlier observations. Although SoLS is provided with a history of actions, a history of observations is not, due to context-length restrictions and concerns about information overload. In addition to memory-related challenges, other tasks might depend on visual input, either directly through images or because the textual representation of certain items, such as mazes, is inadequate. Furthermore, a few tasks involve using phone features like the clipboard, which the agent has never encountered and is unable to intuitively understand. Lastly, some tasks demand very long sequences of interactions, with several requiring 15 to 60 steps for an "optimal" solution. These tasks are inherently difficult, and the likelihood of the agent making a mistake, particularly without a rich context of past observations, increases significantly. These failure cases provide avenues for future work, such as incorporating a better form of memory into agents alongside SoLS, or using base models with visual input and broader fine-tuning data.

# 5   Related Work

## 5.1   Reinforcement Learning with Foundation Models

There has been a growing focus on applying a final RL step to LLMs to improve alignment with human values, a process known as RL from Human Feedback (RLHF) [7, 40]. In RLHF, a reward model is trained on human-labelled data, then the LLM is trained to maximise predicted rewards. To prevent "reward hacking" [10, 13, 18], the policy is regularised using algorithms like PPO [29]. RL is also used to fine-tune models for specialised tasks like math and coding [16, 30].

Several works explored LLMs for multi-turn tasks. Abdulhai et al. [1] created tasks using the OpenAI Gym interface to evaluate various RL algorithms. While numerous studies apply RL to multi-turn

tasks [4, 8, 39], many focus on small-scale environments such as AlfWorld [31] or BabyAI [6]. Our work aims to apply RL with LLM-based policies in real-world environments, where tasks are more complex and simulation speeds are slow. DigiRL [2] attempts online RL but mostly evaluates on tasks with small distribution shift between SFT and RL steps.

Recently, there has been a turn towards critic-free RL algorithms. Following DeepSeek-R1's success [16], several works use Group Relative Policy Optimization (GRPO) [30], which estimates the advantage function by generating multiple responses per prompt and using their average reward as baseline. Extensions have been proposed [5, 24] to improve GRPO. However, GRPO is expensive, requiring typically 64 responses per prompt, while our SoLS generates a single response per prompt.

## 5.2 Mobile App Agents

Many recent works focus on mobile app agents for Android smartphones. Most current agents rely on prompting large proprietary models like GPT-4o. Complex prompting agents achieve strong performance through exploration, planning, and reflection [32, 34, 37], sometimes with replanning and multiple passes [34]. However, this intricate prompting requires numerous costly API calls.

Some agents use two-step prompting methods, planning-grounding [38] or acting-summarising [28]. In subsequent work, agents were developed with fine-tuned grounding models integrated into the pipeline, combining GPT-4o and fine-tuned models into planning-grounding-summarising [14, 36]. While achieving some of the best online app control results to date, they remain slow and costly, requiring two proprietary API calls per step. Other agents explored using exclusively fine-tuned models [9, 20, 21, 23, 27, 35]. Several employ two-stage training, using GUI grounding datasets prior to action generation fine-tuning [21, 23, 35]. However, fine-tuned models struggle to match large proprietary LLMs' performance, and some works are limited to offline evaluation [9, 20, 35].

DigiRL [2] is the most closely related, using SFT followed by RL fine-tuning for mobile app control. They performed SFT on a VLM using small AitW subsets, then conducted RL on similar tasks. Our evaluation differs significantly by focusing on RL in tasks that are OOD compared to the initial SFT dataset, substantially increasing difficulty.

## 6 Conclusion

In this paper, we introduced Succeed or Learn Slowly (SoLS), a novel off-policy RL algorithm for mobile app control tasks. SoLS addresses the challenges of sparse rewards and high simulation costs through asymmetric policy updates. The core innovation is enabling aggressive learning from successful experiences while applying conservative regularisation to unsuccessful ones. Combined with Successful Transition Replay (STR), SoLS achieves a 51.3% overall success rate on AndroidWorld, outperforming both state-of-the-art GPT-4o-based and alternative RL methods. Moreover, this is done with 5-60x faster inference time than the best approaches.

Despite its strong empirical performance, SoLS has several limitations. The asymmetric constraint mechanism can potentially lead to premature convergence to local optima, especially when actions receive positive advantages early in training due to noise or limited exploration. SoLS also depends heavily on the quality of the initial SFT policy; if that policy has significant deficiencies in certain action types or task domains, SoLS may struggle due to its conservative updates for negative-advantage actions. In highly stochastic environments, where the same action may succeed or fail inconsistently, SoLS can suffer from inconsistent feedback, potentially causing policy oscillations. Lastly, although STR aids knowledge transfer, SoLS fundamentally cannot learn to succeed in tasks requiring interaction patterns entirely absent from both the initial policy and exploration distribution.

While this work focuses on technical advancements, the development of mobile app control agents raises important societal concerns as these agents develop further. Such systems could eventually be used to compromise user privacy through automated data access, enable new forms of digital surveillance, and create security vulnerabilities if misused for unauthorised access to personal devices.

Our work demonstrates that well-designed RL algorithms can enable smaller language models to outperform much larger foundation models on specialised tasks, with important implications for resource-constrained environments. Future work could explore shaped rewards, improved exploration strategies, and theoretical properties of selective constraint mechanisms.

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

# A  Experimental Details

## A.1  Notation

Table 2: Notation used in equations.

| Symbol | Description |
|---|---|
| $a$ | Action taken by the agent |
| $s$ | Environment state |
| $r$ | Episode-terminal reward |
| $\epsilon$ | Clipping parameter controlling trust region around importance ratio |
| $R$ | Monte Carlo return |
| $A$ | Advantage function |
| $V^{\pi_\theta}$ | Value function under policy $\pi_\theta$ |
| $Q^{\pi_\theta}$ | Action-value function under policy $\pi_\theta$ |
| $\pi_\theta$ | Parametric policy with parameters $\theta$ |
| $\pi_b$ | Behavioural policy used to generate training data |
| $d^{\pi_b}(s)$ | State distribution under policy $\pi_b$ |
| $\mathcal{D}$ | Training dataset |
| $\mathcal{D}_{\mathrm{STR}}$ | STR dataset |
| $\mathcal{D}_{on}$ | On-policy dataset |
| $\mathcal{D}_{off}$ | Off-policy dataset |
| $\mathcal{J}(\theta)$ | Objective function of policy parameters $\theta$ |
| $\mathcal{L}_{ac}$ | Actor loss function |
| $\mathcal{L}_{cr}$ | Critic (value regression) loss |
| $\mathcal{L}$ | Combined actor-critic SoLS loss function |

## A.2  Implementation Details

We use Llama-8B-Instruct as our base model. We fine-tune the model using LoRA [17] adapters to minimise computational requirements. The AdamW optimiser [25] is used across all experiments.

During SFT, we update the model parameters for three epochs using the AndroidControl training set. We apply a learning rate of $10^{-4}$ that linearly decays to 0. We use an effective batch size of $64$. The LoRA adapters are configured with $64$ dimensions, lora-$\alpha$ of $32$, and dropout rate of $0.05$.

During RL fine-tuning, we train the model for 15K episodes, equivalent to approximately 200K transitions. This transition count varies between algorithms since those with higher success rates complete episodes more quickly, resulting in fewer overall transitions. For training, we implement data parallelism with 8 concurrent processes. Each process asynchronously interacts with emulators and collects data until accumulating 100 on-policy transitions. These 100 timesteps, combined with 50 transitions sampled from the STR, are used for learning. We perform mini-batch gradient updates with batch size $64$ and set the $\epsilon$ hyperparameter to $0.2$. The learning rate for RL fine-tuning remains constant at $10^{-5}$ throughout training. We conduct a single training epoch for all algorithms except PPO, for which we perform two epochs on the training data.

For value function estimation, we add an extra affine value head on top of the policy's last hidden layer. We allow value loss gradients to propagate through all trainable model parameters. To balance the policy and value loss functions, we set $\lambda$ to $0.5$.

## A.3  DigiRL Details

In our implementation of DigiRL-STR we use the STR to augment the training algorithm and not the original prioritised experience replay used by the authors. We used this modification to ensure consistency among training algorithms. DigiRL also performs step-level filtering, where each transition is filtered using a step-based advantage estimation (Equation 4.3 of the original paper), presented below and adjusted to our notations:

$$A^{\text{step}}(s_t, a_t) = \lambda^{H-t} r(s_H, a_H) + (1 - \lambda^{H-t} r(s_t, a_t))(V^{\text{step}}(s_{t+1}) + r(s_t, a_t, c) - V^{\text{step}}(s_t)) \quad (9)$$

where $H$ is the horizon, and the value of $\lambda$ is set to 0.5. Transitions with advantage larger than 0.05 are used for training, and the rest are discarded. Note that even though DigiRL is off-policy, it does not use importance sampling, instead opting for other measures. We keep this consistent with the original algorithm.

### A.4 Data and Benchmark

#### A.4.1 Action and observation space

In this section, we describe the action space and observation inputs used by our SoLS agent, as well as other RL methods. For the remaining baselines, we adopt their respective action spaces and observation formats, using provided code with only minor modifications where applicable.

The action space of our agent adheres to a standardised JSON-style format, specifying both the action type and any optional parameters: `{"action-type":<type>, "action-extra":<extra>}`. The list of available `action-type` and `action-extra` options is detailed in Table 3, and is consistent across both the AndroidControl and AndroidWorld environments. Standardising the action space across the SFT dataset and the RL environment is critical to ensure effective transfer of training. Therefore, actions are consistently converted to and from this format during training and evaluation in both environments. As in prior work [e.g., 9, 20, 28, 38], the `click` and `long-press` actions operate on target UI elements rather than on explicit $(x, y)$ screen coordinates. This design choice simplifies action generation and increases the likelihood of generating robust, executable actions, especially in the absence of GUI-grounding training. The referenced target element can be translated into an $(x, y)$ coordinate at execution time using its bounding box.

| Action type | Action extras |
|---|---|
| open-app | \<app name\> |
| input-text | \<text\> |
| click | \<target element\> |
| long-press | \<target element\> |
| wait | - |
| scroll-up | - |
| scroll-down | - |
| scroll-left | - |
| scroll-right | - |
| navigate-home | - |
| navigate-back | - |

Table 3: Our action space.

Figure 4: Input to SoLS and other RL methods.

As described in Sections 2.2 and 3.1, both AndroidControl and AndroidWorld provide the task goal, the UI accessibility tree, and a screenshot of the phone interface at each step. Since SoLS and the other RL baselines we evaluate are based on Llama-3-8B and are text-only models, the screenshot is omitted. This significantly reduces the number of input tokens, thereby decreasing the model's inference time. The UI tree is transformed into a list of UI elements, each represented in a JSON-like format that includes any relevant text or metadata provided by the tree. This is concatenated with the textual goal description to form the first part of the model's input. The final part of the input is the history of actions, which is recorded as the agent progresses through each task and is appended to the model's input. A visualisation of the overall observation and input structure is shown in Figure 4.

#### A.4.2 Evaluation and benchmark set details

As discussed in Section 2.2, the AndroidWorld benchmark is used for evaluation throughout our experiments. Although the original benchmark consists of 116 tasks, we employ a subset of 80 tasks. Q&A tasks are excluded, as they represent a distinct category not supported by our agents, action space, or by AndroidControl. Verification tasks are also removed, since our evaluation procedure checks for task success at every step, rendering these tasks trivially solvable. Lastly, we exclude tasks that require free-form actions, such as drawing, as these are incompatible with the agents' current action spaces. Notably, most of the omitted tasks fall under the "easy" difficulty category, resulting in a task distribution that is more challenging than that of the full benchmark suite, as shown by the task distribution charts in Figure 5.

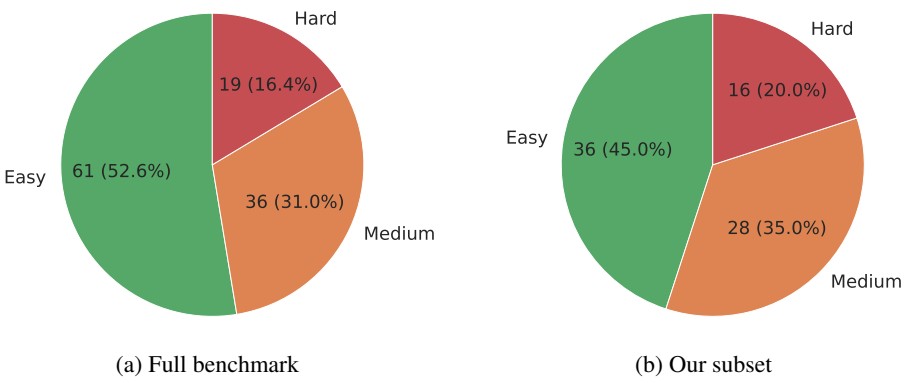

(a) Full benchmark                    (b) Our subset

Figure 5: Pie charts comparing task difficulty distribution between the full AndroidWorld benchmark, and the task subset used in this work.

# B    Additional Results

## B.1    Inference Time and Model Size

Figure 2b illustrates the trade-off between success rate and inference time across different agents, demonstrating that SoLS not only outperforms the baselines in terms of success rate but also achieves significantly faster inference times. In Figure 6, we further examine the model sizes and inference times of the various agents side-by-side, offering insight into the memory, compute, and latency requirements of each approach. Purely fine-tuned methods, such as OS-Atlas-Pro and our RL implementations, are the least demanding in terms of both model size and inference time. Note that we report the model size of AriaUI as 24.9B, reflecting the total number of parameters, even though only 3.9B are active during inference.

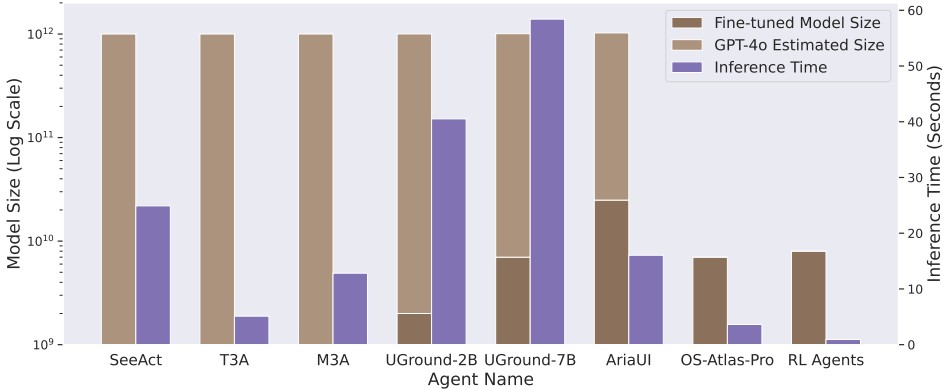

Figure 6: Model size and inference time of different agents. GPT-4o size is estimated at 1 trillion parameters and is shown on top of the fine-tuned grounding model size for mixed prompting-fine-tuned agents.

## B.2    Failure Modes and Case Studies

The main failure modes and an analysis of the types of tasks that SoLS repeatedly fails to solve are introduced in Section 4.4. This section provides further explanation, examples, and illustrations of these failure cases. A representative example for each failure category is included in Table 4.

**Lack of memory.**    Some tasks require the agent to retain information across multiple steps. While SoLS receives a history of past actions as part of its input, this provides only limited information about prior states. Due to context-length limitations and computational considerations, the history of

Table 4: Example failed tasks for each failure category.

| Failure Mode | Example |
| --- | --- |
| Memory | Open the file task.html in Downloads in the file manager; when prompted open it with Chrome. Then click the button 5 times, remember the numbers displayed, and enter their product in the form. |
| Visual Input | Add the recipes from recipes.jpg in Simple Gallery Pro to the Broccoli recipe app. |
| Unseen Interactions | Copy the following text to the clipboard: {clipboard_content} |
| Long Tasks | Save a track with waypoints Ruggell, Liechtenstein, Bendern, Liechtenstein in the OsmAnd maps app in the same order as listed. |

observations is not included. Consequently, SoLS struggles with tasks that depend on remembering past information, as illustrated by the example in Table 4. A possible mitigation strategy in future work is the integration of a state summarisation mechanism at each step, which can be fed into subsequent model inputs. A common approach in related works [e.g., 14, 28, 36] is to use GPT-4o as an external summarisation module. However, this incurs the cost of querying GPT-4o at every step.

**Lack of visual input.**  SoLS relies solely on text-based inputs, which leads to two primary challenges. First, although UI accessibility trees typically provide high-quality textual representations of elements, they can be inadequate for image-based elements. This is exemplified in Figure 7a, where the maze's layout and the agent's position are not described textually, leaving the agent to take essentially random actions. Second, some tasks require extracting information from images, capabilities that a text-only model cannot support. One such case is shown in Table 4. Mitigating this failure mode may require transitioning to a Vision-Language Model (VLM). Alternatively, one could apply Optical Character Recognition (OCR) to extract textual information from image components of the UI element list and incorporate it into the input.

**Unseen interactions.**  Tasks in AndroidWorld are out-of-distribution (OOD) relative to the AndroidControl SFT data. Although many tasks can be addressed by generalising to new applications through similar interaction patterns, some require the use of previously unseen device features or unfamiliar actions. For instance, the clipboard functionality, illustrated in Table 4, has never been encountered by SoLS. Given the rarity of the `long-press` action in the training data (only 0.2%), the agent is unlikely to discover and reinforce correct usage through exploration. A similar issue is presented in Figure 7b, where the agent attempts to append text to a pre-filled field without first clearing it, again requiring a rarely used `long-press` action. A potential mitigation is to expand the SFT dataset to include a broader variety of tasks and interactions, or to inject exploratory behaviours from other models during RL training.

**Long-horizon tasks.**  Certain tasks in AndroidWorld involve lengthy action sequences. Specifically, nine tasks have optimal trajectories exceeding 20 steps, and three exceed 30. One particularly difficult example in Table 4 requires a 60-step optimal solution. Such tasks are inherently difficult for any agent, as the probability of making a critical error increases with sequence length. This is especially problematic in sparse reward settings, where the agent only receives a positive reward upon full task completion. In such scenarios, RL agents can only reinforce successful strategies after discovering a full solution, which is highly improbable given the length of these tasks. While difficult to fully mitigate, incorporating memory mechanisms (as discussed in the first failure mode) could help improve performance on these long-horizon tasks.

**Combined failure modes.**  Many of the most challenging tasks fall under multiple failure categories simultaneously. This is particularly true for tasks involving visual input, which often also require memory to retain image-derived information over multiple steps. In fact, many such tasks are additionally long-horizon, making them exceptionally difficult under current system limitations.

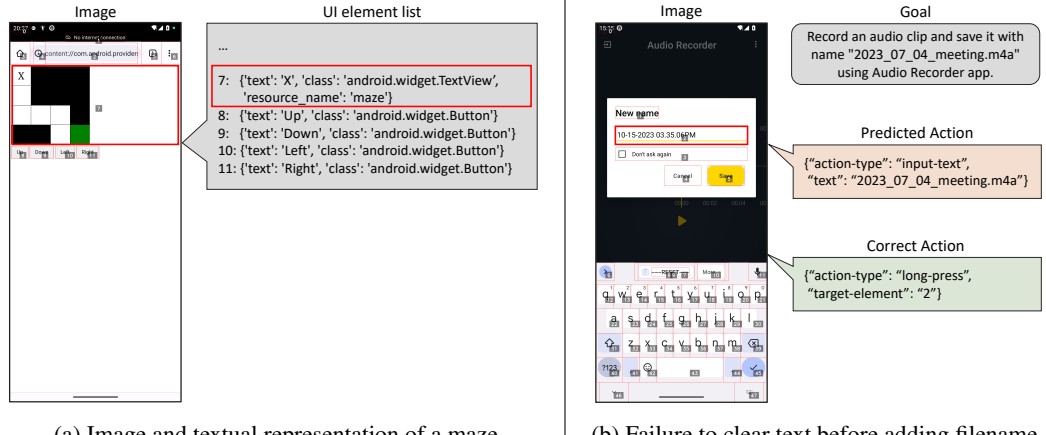

| (a) Image and textual representation of a maze | (b) Failure to clear text before adding filename |

Figure 7: Two examples illustrating the cause of failure cases. (*left*) The textual representation of the maze, highlighted in red, does not describe the content of the maze visible in the image. (*right*) Having never seen pre-filled text fields, the agent tries to input the filename, instead of trying to clear it first, with the `long-press` action.

## B.3 Training Curve

Figure 8 presents the performance of a training run of SoLS-STR, as success rate across episodes. Final training success rate is slightly lower than the evaluation value in Table 1, due to higher sampling temperatures used during training.

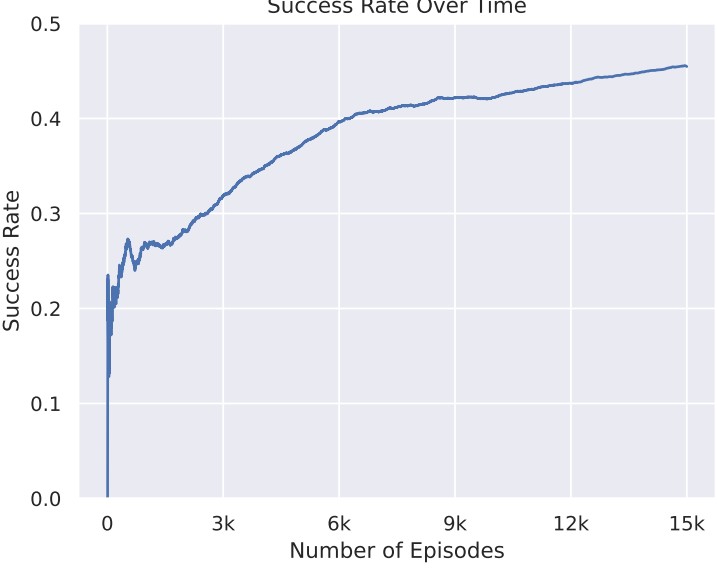

Figure 8: SoLS-STR success rate throughout training.

