# OpenReview forum: "Succeed or Learn Slowly: Sample Efficient Off-Policy Reinforcement Learning for Mobile App Control"
_NeurIPS.cc/2025/Conference — NeurIPS 2025 poster_

### Official Review · Reviewer_Css9 · 2025-06-02

**Clarity:** 4
**Significance:** 4
**Originality:** 4
**Rating:** 5
**Confidence:** 1

**Summary:**

10373

Summary:

This paper considers the use of RL to train agents for mobile application interaction. It recognizes two challenges: dealing with sparse rewards, and dealing with limited samples. It addresses these challenges with two innovations. The first is an alteration to PPO-like algorithm that favors the reinforcement of successful actions, while more strongly regularizing (preventing significant policy changes) for unsuccessful actions. [Note that trajectory have binary rewards, and so a successful actions is one that results in the trajectory having a reward of 1.] The authors claim that this helps prevent catastrophic forgetting during exploration. The second innovation is a variant of experience replay, which the authors call Successful Transition Replay (STR), which prioritizes successful state-action pairs from previous episodes.

The authors evaluate their method on a benchmark called AndroidWorld, obtaining an overall success rate (across three problem difficulties) of 51.3% (17% above the next best), significantly outperforming GPT-4o-based approaches  and other methods based on RL fine-tuning. Their method also achieved this with between a 5x and 60x computational speed-up.

**Questions:**

What evidence did you observe to support the claim that the bias towards successful actions reduces catastrophic forgetting?

**Ethical Concerns:**

["NO or VERY MINOR ethics concerns only"]

**Final Justification:**

I have read the author's response, and it has not changed my evaluation. I continue to recommend acceptance.

**Limitations:**

None.

**Paper Formatting Concerns:**

None.

**Quality:**

3

**Strengths And Weaknesses:**

Strengths. The paper is clear and well-written. It clearly describes the problem it sets out to solve, explains the proposed method, and presents empirical results supporting the claims.

Weakness: Although the paper claims that performance improves due to the bias towards successful actions preventing catastrophic forgetting, that claim is not well-supported by the experiments. It's also not clear to my why this would prevent catastrophic forgetting. However, if the paper removed this claim, it would still be showing improved performance on an existing benchmark.

I note, however, that this paper is outside my area. I am not familiar with any of these algorithms (other than PPO) nor benchmarks. That is a statement about my expertise, not the choices of algorithms / benchmarks in the paper.

---

> ### Author Rebuttal · Authors · 2025-07-30
>
> We thank the reviewer for their valuable feedback and comments. We address the main concerns below and clarify key points raised in the review.
>
> **Weaknesses / Questions:**
>
> > Although the paper claims that performance improves due to the bias towards successful actions preventing catastrophic forgetting, that claim is not well-supported by the experiments. It's also not clear to my why this would prevent catastrophic forgetting. However, if the paper removed this claim, it would still be showing improved performance on an existing benchmark.
> What evidence did you observe to support the claim that the bias towards successful actions reduces catastrophic forgetting?
>
> We acknowledge that our claim regarding “catastrophic forgetting” could have been more clearly explained and better supported. The main evidence we intended to highlight is the comparison with A2C-STR, which does not apply conservative updates to negative-advantage actions. In practice, A2C-STR exhibited training instability and required significant tuning effort, whereas SoLS-STR did not and achieved a 60% relative improvement in final performance (51.3% vs 32.1%).
>
> That said, we agree that “catastrophic forgetting” may not be the most precise term for this phenomenon. A more accurate framing would be that SoLS helps prevent performance degradation due to negative updates during RL fine-tuning. We will rephrase this claim accordingly in the next revision to better reflect the observed behaviour.

---

### Official Review · Reviewer_j81U · 2025-06-29

**Clarity:** 3
**Significance:** 3
**Originality:** 3
**Rating:** 5
**Confidence:** 4

**Summary:**

The paper proposes SoLS, an off-policy RL algorithm for efficient fine-tuning of language models in mobile app control. It uses asymmetric policy updates and introduces STR to prioritize learning from successful steps. On the AndroidWorld benchmark, SoLS outperforms GPT-4o-based and prior RL methods.

**Questions:**

1. Can you clarify how STR handles tasks with very few or no successful transitions early on? Is there a cold-start issue, and if so, how does the method mitigate it?
2. Given that SoLS suppresses updates for negative-advantage actions, how does the model learn to handle rare or unseen actions (e.g., long-press or clipboard tasks)? Is there a risk of overfitting to early successes?
3. Do you think SoLS could generalize to other structured control settings beyond mobile apps—such as form-filling on websites or desktop UI automation? (Not a request for additional experiments, just curious about your perspective.)
4. Can you quantify the total compute cost of SoLS training (e.g., GPU-hours)? This would help assess the practical trade-offs between your method and API-heavy approaches like GPT-4o.
5. How sensitive is SoLS to the choice of $\epsilon$ in the clipped importance sampling update? Did you experiment with different values, and if so, how stable was training across them?

**Ethical Concerns:**

["NO or VERY MINOR ethics concerns only"]

**Limitations:**

Yes.

**Paper Formatting Concerns:**

No.

**Quality:**

3

**Strengths And Weaknesses:**

**Strengths**

1. The method works well even with a relatively small model. This has clear implications for cost-effective deployment.

2. The writing is clear, and the motivation and methodology are easy to follow. The authors also include a thoughtful discussion of limitations and failure cases, which strengthens the paper.
3. The results are strong. SoLS beats all the baselines, including GPT-4o-based agents, and does so while being way faster at inference.



**Weaknesses**

1. The method is quite tailored to mobile app environments, where the action space is structured and discrete. It’s unclear how well SoLS would generalize to other domains with less constrained action formats (e.g., open-ended text generation or robotics)
2. The success of SoLS seems to depend heavily on the quality of the initial supervised fine-tuning. If the SFT model is poor or underexposed to certain action types, SoLS may not recover well, since negative updates are largely suppressed.
3. STR only stores successful timesteps, not full trajectories or multi-step episodes. This design improves stability and efficiency but might miss longer-term dependencies or subgoal information, which could be important for some tasks.

---

> ### Author Rebuttal · Authors · 2025-07-30
>
> We thank the reviewer for their valuable feedback and comments. We address the main concerns below and clarify key points raised in the review.
>
> **Weaknesses:**
>
> 1. > The method is quite tailored to mobile app environments, where the action space is structured and discrete. It’s unclear how well SoLS would generalize to other domains with less constrained action formats (e.g., open-ended text generation or robotics)
>
> SoLS is specifically designed for environments with structured, discrete action spaces, where actions follow a well-defined format. This makes SoLS well-suited not only to mobile UI environments but also potentially to other domains with similar constraints, such as web UI and robotics, where action spaces are often structured and bounded.
> In contrast, open-ended domains like free-form text generation present different challenges. In such settings, policy regularisation remains critical to prevent reward hacking and maintain generation quality, as supported by the literature on PPO and GRPO. SoLS’s selective update mechanism may be less appropriate in these domains without significant adaptation.
> In summary, while SoLS is not intended as a universal RL framework, we see it as a step toward scalable and stable RL with foundation models in high-cost, structured environments. We hope future work can extend its principles to less constrained settings.
>
>
> 2. > The success of SoLS seems to depend heavily on the quality of the initial supervised fine-tuning. If the SFT model is poor or underexposed to certain action types, SoLS may not recover well, since negative updates are largely suppressed.
>
> We agree with the reviewer that SoLS relies on a reasonably competent initial SFT model to be effective. This is a deliberate design trade-off: by applying conservative updates to negative-advantage samples, SoLS avoids harmful drift but limits its ability to recover from a poor starting point.
>
> That said, this reliance is not unique to SoLS, but rather reflects a fundamental challenge in RL with large action spaces, especially in the LLM setting. Unlike traditional RL settings with small discrete action spaces (e.g., 5 actions), foundation models and mobile phones operate over several actions, making it infeasible to learn effectively through random exploration alone. Without a strong SFT phase to anchor the initial policy, any RL method would struggle to discover successful actions purely via exploration.
> In our work, the combination of SoLS with STR helps address this by focusing updates on rare but informative successes, improving sample efficiency. Nevertheless, we acknowledge this as a limitation and an important direction for future work.
>
>
> 3. > STR only stores successful timesteps, not full trajectories or multi-step episodes. This design improves stability and efficiency but might miss longer-term dependencies or subgoal information, which could be important for some tasks.
>
> This is simply a design choice of the algorithm, because a timestep is self-sufficient. In practice, the data is kept in steps, as each step contains all necessary information, including current observation, past actions, and return, to compute the loss function of SoLS. It could be easily modified to store trajectories if they were needed, for example, to compute the advantage based on GAE.
>
>
> **Questions:**
> 1. > Can you clarify how STR handles tasks with very few or no successful transitions early on? Is there a cold-start issue, and if so, how does the method mitigate it?
>
> Indeed, tasks with few or no early successes may face a cold-start issue. However, SoLS is designed to mitigate this in two key ways:
> On-policy data is always included in every update batch, alongside sampled STR data (see Equation 2). This ensures that learning continues even for tasks without prior successful transitions, allowing the policy to gradually improve through exploration.
> The initial SFT model provides a reasonable starting point, enabling early successes on a subset of tasks. These early successes help populate the STR buffer and, over time, generalise to harder or less familiar tasks via shared model weights and transferable behaviours.
>
> In practice, we observe that STR grows organically throughout training. While some tasks may have slower start times, the combination of SFT bootstrapping and on-policy rollouts ensures that cold-start tasks are not excluded from learning.
>
>
> 2. > Given that SoLS suppresses updates for negative-advantage actions, how does the model learn to handle rare or unseen actions (e.g., long-press or clipboard tasks)? Is there a risk of overfitting to early successes?
>
> SoLS’s conservative handling of negative-advantage updates does present challenges for learning rare or unseen actions, especially if they are not well-covered during SFT. However, we address this through several design choices:
> Aggressive updates on positive advantages enable the model to rapidly reinforce successful rare actions, even if they occur only once during exploration. This allows the model to quickly adapt once such actions are discovered.
>
> Successful Transition Replay (STR) helps maintain diversity and avoids overfitting to early successes by storing and sampling successful steps on a per-task basis. This ensures the policy continues learning from newer and more varied experiences and prevents drowning out the more difficult tasks, mitigating overfitting to easier tasks.
>
> Regarding easy tasks, we observe that the value function quickly converges, and the advantage for those tasks tends toward zero. As a result, updates to the policy are minimal (since they are scaled by the advantage), preventing overfitting of already-solved behaviours. This stability allows SoLS to focus learning capacity on harder tasks where the advantage signal is still meaningful.
>
>
> 3. > Do you think SoLS could generalize to other structured control settings beyond mobile apps—such as form-filling on websites or desktop UI automation? (Not a request for additional experiments, just curious about your perspective.)
>
> Yes, we believe SoLS could generalise well to other structured control settings beyond mobile apps, such as web form-filling, desktop UI automation, or even robotic task planning. Some current mobile app tasks, such as creating contacts or recipes, actually involve form-filling already. Generalisation to web-based form-filling or desktop UI should be reasonably seamless, as the domains are very similar.
>
> These other settings share the same core challenges that SoLS is specifically designed to address: sparse rewards, large but structured action spaces, and possibly costly interactions. The separation of positive and negative updates, combined with STR, can offer similar benefits in those settings without requiring major algorithmic changes.
>
> We appreciate the reviewer’s perspective and view this as a promising direction for future applications.
>
>
> 4. > Can you quantify the total compute cost of SoLS training (e.g., GPU-hours)? This would help assess the practical trade-offs between your method and API-heavy approaches like GPT-4o.
>
> We used 8 V100 GPUs, where training took approximately 5 days, consistent across RL methods. This could be significantly sped up if the experiments were performed with newer GPUs.
>
>
> 5. > How sensitive is SoLS to the choice of epsilon in the clipped importance sampling update? Did you experiment with different values, and if so, how stable was training across them?
>
> We did not experiment with the epsilon value due to computational constraints and the cost of running training and evaluations. We maintained the standard value of 0.2.

---

> > ### Comment · Reviewer_j81U · 2025-08-02
> > **Thank the authors for the detailed rebuttal!**
> >
> > Thank the authors for the detailed rebuttal. Most of my concerns have been addressed.
> > I understand the computational constraints and agree that ablations on the clipped importance sampling epsilon are not expected at the rebuttal phase. Still, I encourage adding such analysis in future versions.
> > I maintain the rating as "accept" and wish you the best with your submission.

---

### Official Review · Reviewer_nx8y · 2025-07-01

**Clarity:** 3
**Significance:** 3
**Originality:** 3
**Rating:** 4
**Confidence:** 4

**Summary:**

The paper proposes a two-stage approach consisting of supervised fine-tuning on human demonstrations, followed by reinforcement learning with SoLS, which applies direct policy updates for positive advantages and PPO-style KL regularization for negative ones. To improve efficiency, training focuses on selectively replaying successful transitions.

**Questions:**

1. The concept of the behavior policy in Section 3.3 requires further clarification.

2. The authors claim that SoLS is an off-policy method, but the experiments only compare it with on-policy methods. The authors should explain this choice, and if possible, add comparisons between SoLS and state-of-the-art off-policy methods.

3. The explanation of how STR addresses a significant distribution shift from the original SFT phase needs further elaboration.

4. SoLS does not clip updates for positive advantages, unlike PPO, which clips all updates to ensure stability. While this may accelerate learning, it could also lead to instability if the advantages are noisy. Please clarify this design decision and provide supporting evidence.

5. The idea of selectively updating the policy based on chosen samples seems to have been explored in the DigiRL. The main difference appears to lie in the thresholding strategy. The novelty of SoLS needs to be more clearly articulated.

**Ethical Concerns:**

["NO or VERY MINOR ethics concerns only"]

**Final Justification:**

The rebuttal has successfully clarified the novelty of the SoLS method, the effectiveness of STR in mitigating distribution shift, and the details of the experimental setup. These clarifications have effectively addressed many of my initial concerns, particularly with respect to the method’s originality and the rigor of the evaluation. Overall, the paper demonstrates sufficient innovation and empirical strength. However, I recommend that the score update be made contingent upon the authors addressing two remaining issues and providing further clarifications in the final published version.

**Limitations:**

Yes

**Paper Formatting Concerns:**

No major formatting issues found.

**Quality:**

3

**Strengths And Weaknesses:**

Strengths:
1.Successful Transition Replay(STR) improves sample efficiency by selectively replaying successful transitions.
2.The proposed SoLS applies direct policy updates for positive advantages and conservative, regularized updates for negative ones, reducing policy degradation.

Weaknesses:
1.There are instances of unclear descriptions. For example, the term "behavior policy" in Section 3.3 needs further clarification.
2.The terms in equations should be explicitly defined and clarified.
3.The acronym "RLHF" appears for the first time in Section 3.3 without being spelled out.

---

> ### Author Rebuttal · Authors · 2025-07-30
>
> We thank the reviewer for their valuable feedback and comments. We address the main concerns below and clarify key points raised in the review.
>
> **Weaknesses:**
> 1. > There are instances of unclear descriptions. For example, the term "behavior policy" in Section 3.3 needs further clarification.
>
> The “behavioural policy” refers to the policy that was used to generate the data contained in the training batch. In off-policy reinforcement learning, the agent learns about a different policy (the target policy) than the one that generated the data. We will clarify where this term is first used, with something along the lines of “that is, the policy that was used to generate the training data, which may differ from the policy currently being optimised.”
>
> 2. > The terms in equations should be explicitly defined and clarified.
>
> We thank the reviewer for this feedback. We agree that several equations, particularly in Section 3, would benefit from more explicit definitions of all terms. In the next version, we will revise the equations to clearly define each variable and function upon first use or within a terminology table, including notations like $d^{\pi_b}(s)$, $Q^{\pi_\theta}(s, a)$, and the advantage function $A(s, a)$. We will also ensure that all assumptions (e.g., use of Monte Carlo returns, importance sampling ratios, clipping thresholds) are clearly stated in context. We will also better define terms such as the behavioural policy, as stated above.
>
> 3. > The acronym "RLHF" appears for the first time in Section 3.3 without being spelled out.
>
> Apologies for this, we moved the related works section towards the end of the paper at the last minute. We will change this so that the first use of RLHF spells out the full acronym.
>
> **Questions:**
> 1. > The concept of the behavior policy in Section 3.3 requires further clarification.
>
> We kindly refer the reviewer to Weakness 1 above.
>
> 2. > The authors claim that SoLS is an off-policy method, but the experiments only compare it with on-policy methods. The authors should explain this choice, and if possible, add comparisons between SoLS and state-of-the-art off-policy methods.
>
> SoLS is off-policy in the sense that it uses data from STR (collected under previous policies) alongside on-policy data. It also performs updates using mini-batches of collected data, similar to standard PPO implementations. We maintained consistent implementation details across all algorithms to ensure fair comparison.
>
> However, the baseline methods we compare against are off-policy in the same manner. The buffered updates are a necessary component of the training process for foundational models, which all methods follow. Additionally, we systematically explore the use of off-policy STR data across all RL methods in our evaluation. Particularly, we would like to highlight Figure 2a, which illustrates this quite well. In this figure, we compare our method alongside PPO and DigiRL, both with and without off-policy STR data. The results demonstrate that our method outperforms the baselines in both scenarios, while also showing that incorporating STR data yields significant performance improvements across all methods.
>
> 3. > The explanation of how STR addresses a significant distribution shift from the original SFT phase needs further elaboration.
>
> We first recap how the agent faces a significant distribution shift between SFT and RL phases:
>
> - **SFT training:** AndroidControl dataset with wordy task descriptions, different app distribution, and actions like long-press appearing in only 0.2% of examples
> - **RL evaluation:** AndroidWorld with imperative task descriptions, unseen apps, and frequent use of previously rare actions like long-press
>
> This shift causes the SFT model to initially fail at most AndroidWorld tasks (22.1% success rate), making traditional RL exploration inefficient.
>
> STR addresses this in the following ways:
>
> - **Bootstrapping from rare successes:** Even with poor initial performance, the agent occasionally succeeds through random exploration or partial transfer from SFT. STR captures these rare successful experiences that would otherwise be lost.
> - **Accumulating domain-specific knowledge:** As training progresses, STR builds a repository of successful interactions specifically for AndroidWorld tasks. This creates a bridge between the SFT distribution and the target distribution.
> - **Preventing catastrophic forgetting:** Without STR, successful exploration in later training might be forgotten when the agent encounters different tasks. STR preserves these difficult successes across the training process.
>
> Figure 3 shows this effect clearly - categories like Files, Maps, and Markor show dramatic improvement from first to last 1000 episodes, indicating that STR helps the agent learn new task-specific behaviours absent from the SFT distribution. Meanwhile, categories like Audio and System (which transfer well from SFT) show smaller improvements.
>
> The comparison of methods with/without STR (Figure 2a) demonstrates STR's importance: even PPO and DigiRL show substantial improvements when augmented with STR, validating that the technique addresses fundamental challenges in this out-of-distribution setting.
>
> In essence, STR acts as a mechanism to gradually shift the training distribution from the SFT domain toward the target domain by preserving and amplifying successful domain-specific interactions. We will make this clearer in the next revision.
>
> 4. > SoLS does not clip updates for positive advantages, unlike PPO, which clips all updates to ensure stability. While this may accelerate learning, it could also lead to instability if the advantages are noisy. Please clarify this design decision and provide supporting evidence.
>
> Our decision not to clip positive advantage updates is based on the very sparse positive signal of the targeted environment. In mobile app control, positive advantages are extremely rare and valuable. AndroidWorld provides binary terminal rewards, meaning most episodes yield zero reward. When we do observe positive advantages, they represent genuinely successful behaviours that we want to reinforce aggressively without any constraints.
>
> Some supporting evidence:
>
> - **Empirical stability:** Despite applying completely unconstrained updates for positive advantages, we did not observe training instability across our experiments. Our training remained stable throughout.
> - **Comparative performance:** The significant improvement over PPO suggests that aggressive, unconstrained positive updates are beneficial rather than harmful. If instability were a major issue, we would expect more erratic performance.
>
> The fact that even SoLS without STR outperforms PPO (as mentioned in our ablations Section 4.3 - Figure 2a) suggests that the asymmetric clipping itself, rather than just the experience replay, drives improvement.
>
> We acknowledge that in noisier reward environments, this design choice might require more careful consideration, but our experimental evidence supports its effectiveness in structured, sparse reward settings like mobile app control. We will further clarify in the next revision of the paper.
>
> 5. > The idea of selectively updating the policy based on chosen samples seems to have been explored in the DigiRL. The main difference appears to lie in the thresholding strategy. The novelty of SoLS needs to be more clearly articulated.
>
> The observation of the reviewer that both DigiRL and SoLS selectively update based on sample quality is correct, and we will more clearly articulate the key differences in the next revision.
>
> While both methods are selective, they handle negative samples in fundamentally different ways. DigiRL performs complete rejection sampling, where negative advantage samples are entirely discarded and contribute zero learning signal. On the other hand, SoLS performs conservative regularised updates, where negative samples are retained but updated with PPO-style clipping constraints.
> In addition, DigiRL does not weigh the updates with the advantage, while SoLS does, and DigiRL does not perform importance sampling when incorporating off-policy data into its updates.
>
> This matters for a few reasons:
>
> - **Information preservation and sample efficiency:** In sparse reward environments, completely discarding negative samples (as DigiRL does) wastes potentially valuable learning signal. SoLS extracts conservative learning from these samples while preventing harm.
> - **Empirical evidence:** Our empirical results show SoLS-STR (51.3%) significantly outperforms DigiRL-STR (38.8%), a 32% relative improvement. This suggests that conservative learning from negative samples is more effective than complete rejection.
>
> The novelty lies in finding the optimal middle ground between aggressive learning and complete rejection. SoLS introduces asymmetric treatment, with unconstrained aggressive updates for positive advantages, and constrained conservative updates for negative advantages.
>
> This is neither pure rejection sampling nor symmetric regularisation; it's a principled asymmetric approach that maximises learning from both positive and negative experiences while maintaining model stability. We acknowledge that we should better emphasise this key algorithmic distinction from DigiRL, and will do so in the next revision.

---

> ### Comment · Reviewer_nx8y · 2025-08-07
>
> Thank you for your detailed rebuttal. Your responses have successfully clarified the novelty of the SoLS method, the effectiveness of STR in addressing distribution shift, and have resolved my questions regarding the experimental setup.
>
> However, two concerns remain:
>
> 1.The definition of the "behavioural policy" remains unclear.
> While I understand that the behavioural policy refers to the one used to generate the data in off-policy settings, it remains unclear what specific policies are included in "some behavioural policy" as mentioned in Section 3.3. Does this refer to the immediately preceding policy, or a mixture of past policies? If it is the latter, I am concerned that the KL divergence between such a mixture and the current policy could be large, raising the question of whether it is reasonable to constrain policy updates based on their KL divergence.
>
> 2.The claim of training stability lacks sufficient supporting evidence.
> Figure 2a only presents final success rates, which does not directly support the claim regarding training stability. To substantiate this point, it would be helpful if the authors could provide full evaluation curves (e.g., success rate vs. training episodes) to illustrate training dynamics and stability more clearly. Additionally, I am somewhat confused by the statement that positive advantages represent genuinely successful behaviors. Could the authors clarify why a positive advantage at a single step is sufficient to indicate overall success, especially in long-horizon tasks where a local positive advantage may not guarantee eventual task completion?

---

> > ### Author Response · Authors · 2025-08-07
> >
> > Thank you for taking the time to review our rebuttal and provide additional feedback. We appreciate your continued engagement with our work and address your remaining concerns below.
> >
> > 1. Behavioural Policy
> >
> > We appreciate the reviewer highlighting the ambiguity around the behavioural policy referenced in Section 3.3.
> >
> > In our implementation, the behavioural policy $\pi_b​$ refers to the policy used to generate the specific sample being used for training. Since data in our training pipeline is gathered online, the STR buffer gradually accumulates trajectories from different stages of training, resulting in a mixture of past policies over time.
> >
> > In our experiments, the training data comprises a mixture of two distinct sources. The first source consists of data generated by the current policy (the policy we aim to optimise during the update). The second source consists of data stored in the STR. The historical data sampled from the STR contains only positive examples, and no constraints are applied during the training of these samples. Only the data that was generated from the current policy contains negative samples, which are constrained based on SoLS.
> >
> > To summarise, our specific design significantly mitigates the issue of divergence:
> > - **No constraints on historical samples from a mixture of sources:** Since our replay buffer stores only positive examples from past policies, we apply no constraints to this historical data during training. This eliminates the divergence concern for this portion of our training data.
> > - **Constraints only apply to recent samples:** Negative samples can only originate from the immediately preceding policy checkpoint (collected online in the previous round). Our ratio clipping and constraints are applied exclusively to negative examples in the recent samples.
> >
> > 2. (a) Training Stability
> >
> > We acknowledge the reviewer’s concern about insufficient supporting evidence for the training stability. As mentioned in our response to Reviewer WCy2, we recognise that the current manuscript lacks adequate empirical evidence to support strong assertions about stability. We will significantly tone down or remove any such claims in the final version. This was not a primary focus or contribution of our work, and we will ensure the language reflects that, keeping the emphasis on our core technical and empirical contributions.
> >
> > We also plan to include a training plot (success rate vs. training episodes) for our method in the Appendix, to illustrate the training dynamics more clearly across time.
> >
> > 2. (b) Positive Advantage
> >
> > We agree that in general, a single positive advantage step may not guarantee overall task completion, particularly in long-horizon environments. However, in our setting, this signal is more informative due to the nature of the reward structure.
> >
> > Our environment provides only terminal rewards: an episode receives a reward of 1 if the task is completed successfully and 0 otherwise. This means that all intermediate steps in successful trajectories are associated with the same positive return (i.e., the terminal reward), and the resulting advantage estimates reflect this. Therefore, a positive advantage in our setting typically indicates that the step contributed to a successful outcome, as it is derived from the return observed at the end of the episode.
> >
> > While we acknowledge that this doesn’t guarantee every individual step is globally optimal, in sparse-reward environments like ours, advantage-weighted updates on successful episode steps provide a meaningful learning signal. We will clarify this point in the final version of the paper to avoid any confusion.

---

### Official Review · Reviewer_WCy2 · 2025-07-03

**Clarity:** 2
**Significance:** 3
**Originality:** 3
**Rating:** 4
**Confidence:** 3

**Summary:**

This paper addresses the challenge of sample-efficient RL for mobile app control tasks, particularly when fine-tuning foundation models in sparse reward settings. The authors identify that standard RL updates from failed actions can destabilize and degrade a model's performance. They introduce Succeed or Learn Slowly, a novel off-policy actor-critic algorithm. The core idea of SoLS is an asymmetric policy update mechanism: it applies direct, aggressive updates for successful actions, but uses conservative, regularized updates for unsuccessful ones to prevent catastrophic forgetting. To further improve sample efficiency, SoLS is augmented with Successful Transition Replay, a technique that prioritizes learning from a stored buffer of previously successful interactions. Evaluated on the AndroidWorld benchmark, the proposed SoLS-STR method significantly outperforms existing approaches, including those using the much larger GPT-4o model, while also being 5-60x faster at inference.

**Questions:**

1. SoLS vs. Rejection Sampling (DigiRL): Your motivation for SoLS's conservative negative updates is to avoid the destructive policy changes that can occur with standard actor-critic losses. This is similar to the motivation for rejection sampling methods like DigiRL, which avoids this problem by simply ignoring negative-advantage samples entirely. Your results show SoLS-STR dramatically outperforms DigiRL-STR (51.3% vs 38.8% success rate).  Could you elaborate on why "learning slowly" from failures is so much more effective than not learning from them at all in this context? What is the key mechanism that accounts for this significant performance gap?

2. The paper argues that the SoLS algorithm is designed to be more stable than standard RL approaches in this context. However, the authors also acknowledge potential limitations where SoLS could face instability. can you present the training curves of multiple seeds?

**Ethical Concerns:**

["NO or VERY MINOR ethics concerns only"]

**Final Justification:**

The rebuttal successfully clarified that the novelty of SoLS lies in the specific and effective composition of existing RL concepts for the mobile app control domain. In summary, the paper's empirical results are strong enough. However, this updated score is conditional upon addressing the two points above to ensure the scientific integrity and clarity of the final publication.

**Limitations:**

yes

**Quality:**

3

**Strengths And Weaknesses:**

Strengths:
The paper's central insight—that updates from negative samples should be handled differently from positive ones in this domain—is both simple and effective.

Weaknesses:
While the overall SoLS algorithm is novel, its core components are inspired by existing concepts. STR is a specialized form of experience replay , and the conservative update mechanism is conceptually related to the clipping in PPO or the filtering in rejection sampling methods like DigiRL.

---

> ### Author Rebuttal · Authors · 2025-07-30
>
> We thank the reviewer for their valuable feedback and comments. We address the main concerns below and clarify key points raised in the review.
>
> **Weaknesses:**
>
> > While the overall SoLS algorithm is novel, its core components are inspired by existing concepts. STR is a specialized form of experience replay , and the conservative update mechanism is conceptually related to the clipping in PPO or the filtering in rejection sampling methods like DigiRL.
>
> While SoLS draws on established ideas such as experience replay and advantage-based updates, its contributions lie in how it adapts and composes these for the unique challenges of mobile app control with foundation models.
>
> While STR shares the general principle of reusing past experiences, it differs fundamentally from standard replay by storing only successful timesteps rather than all experiences, which is crucial in sparse reward environments where the vast majority of experiences are failures. It also stores individual transitions and uses task-specific buffers for a more balanced task distribution.
>
> Our asymmetric update mechanism goes beyond PPO and DigiRL by:
>
> - Applying importance sampling in off-policy data.
> - Conservatively updating on negative advantages instead of discarding them, as in DigiRL. The difference with DigiRL is further elaborated in the following section.
> - Using an advantage-based policy gradient update, in contrast to DigiRL which only maximises the log likelihood.
>
> These design choices enable SoLS to outperform all baselines, achieving >32% relative improvement over DigiRL and PPO. Our contribution lies in finding the right balance between aggressive learning from positive experiences and conservative but non-zero learning from negative experiences, combined with task-specific experience replay, a combination that proves more effective than existing approaches in our challenging mobile app control setting.
>
> **Questions:**
> 1. > SoLS vs. Rejection Sampling (DigiRL): Your motivation for SoLS's conservative negative updates is to avoid the destructive policy changes that can occur with standard actor-critic losses. This is similar to the motivation for rejection sampling methods like DigiRL, which avoids this problem by simply ignoring negative-advantage samples entirely. Your results show SoLS-STR dramatically outperforms DigiRL-STR (51.3% vs 38.8% success rate). Could you elaborate on why "learning slowly" from failures is so much more effective than not learning from them at all in this context? What is the key mechanism that accounts for this significant performance gap?
>
> SoLS has a few important differences compared to DigiR, which we will clarify in the next revision:
>
> - **Information preservation from negative samples:** Even negative-advantage samples contain valuable learning signal that complete rejection wastes. SoLS retains and constrains them, while preserving learning potential. This also improves sample efficiency in a data-scarce environment.
>
> - **Advantage-weighted updates:** DigiRL treats all positive samples equally, using standard -log π(a|s) loss regardless of how successful the action was. SoLS weighs positive updates by the advantage magnitude A(s,a), creating stronger gradients for highly successful actions and weaker ones for marginally positive actions. This provides more nuanced learning that better reflects action quality.
>
> - SOLS uses importance sampling to weigh each sample, which appropriately handles off-policy data.
>
> Note that to ensure a consistent and controlled comparison across methods, we re-implemented DigiRL within our training framework. This required some minor adjustments to match our shared setup. For example, we did not include the instruction-value replay buffer (Section 4.3 in DigiRL), instead using STR (lines 230-235).
>
> While these changes were made for practical reasons, we do not believe they account for the performance gap. The core training behaviour, retaining only positive-advantage samples and applying uniform log-likelihood updates, remains consistent with the original DigiRL approach. Moreover, in our setup, we observed that the modified step advantage estimator and filtering logic had similar practical effects to SoLS’s advantage-based filtering: positive transitions were retained, and negative ones excluded.
>
> We believe the observed performance gains are primarily attributable to SoLS’s advantage-weighted updates, off-policy correction, and conservative treatment of negative samples, rather than differences in auxiliary components.
>
>
> 2. > The paper argues that the SoLS algorithm is designed to be more stable than standard RL approaches in this context. However, the authors also acknowledge potential limitations where SoLS could face instability. can you present the training curves of multiple seeds?
>
> We appreciate the reviewer’s interest in the stability of SoLS and agree that multi-seed training curves would provide additional empirical support. Unfortunately, due to the high computational cost of training in the AndroidWorld environment (each run requiring thousands of slow emulator interactions), running multiple full training seeds is not feasible within the rebuttal timeline.
>
> That said, we have taken several steps to address stability concerns:
>
> - In Table 1, we report the mean and two standard errors over three independent evaluation runs, providing a partial measure of variance across different task initialisations and evaluation conditions.
> - As mentioned in Section 4.2, during development, SoLS exhibited notably more stable training than baselines. A2C-STR required considerable effort to stabilise and frequently diverged, while SoLS trained consistently across our runs and required minimal tuning.
> - In Figure 2a (STR ablation), SoLS outperforms PPO and DigiRL both with and without STR, indicating that its benefits are not overly sensitive to specific replay buffer configurations and suggesting a degree of robustness in training dynamics. Indeed, this suggests robustness to different training runs and variations.
>
> Finally, we acknowledge the concern raised in our limitations section that unrestricted positive updates may lead to convergence toward local optima, such as focusing too narrowly on simpler tasks. However, this is also dependent on other factors such as the quality of the initial SFT model, the representational capacity of the backbone, and the distribution of tasks in the environment. Importantly, we observed that for simpler tasks, solved early in training, the advantage tends to converge toward zero, effectively suppressing further updates to those actions. This self-regulating dynamic helps prevent overfitting to easy tasks and allows SoLS to shift focus to more complex behaviours over time. While we include this point in the limitations for completeness, we did not observe it to pose a major issue in our experiments.

---

> > ### Comment · Reviewer_WCy2 · 2025-08-05
> > **Response to Authors**
> >
> > Thank you for your detailed rebuttal. Your rebuttal successfully clarified that the novelty of SoLS lies in the specific and effective composition of existing RL concepts for the mobile app control domain. However, two major concerns remain, one of which was introduced by the rebuttal itself.
> >
> > 1. Unsubstantiated Claims of Training Stability:
> >
> > A key motivation for SoLS is its purported training stability compared to baselines like A2C. While I understand the computational expense of running multiple seeds in the AndroidWorld environment, the claim of improved stability remains anecdotal without direct empirical evidence. The standard errors reported in Table 1 reflect evaluation variance on a single trained model, not the stability of the training process across different random seeds. To substantiate this claim, you must either:
> >
> > Provide training curves (e.g., success rate vs. episodes) for at least 2-3 random seeds to offer direct evidence of stability.
> >
> > Significantly tone down or remove the claims of enhanced training stability in the final version of the paper.
> >
> > 2. Critical Ambiguity in the DigiRL-STR Baseline Implementation:
> >
> > The most critical concern is the methodological ambiguity surrounding the implementation of the DigiRL-STR baseline. Your paper states that SoLS is an off-policy algorithm that uses importance sampling to correct for data drawn from the STR buffer. You also correctly applied this correction to the A2C-STR baseline. However, your description of the re-implemented DigiRL-STR baseline does not mention whether this same crucial off-policy correction was applied.
> >
> > DigiRL is a form of rejection sampling or filtered behavioral cloning. When applied to off-policy data from the STR buffer, it also requires importance sampling to ensure a fair comparison and avoid biased, high-variance gradient estimates that can cripple performance. If importance sampling was omitted from the DigiRL-STR baseline, the reported 12.5% absolute performance gap between SoLS-STR and DigiRL-STR could be an artifact of a methodologically flawed comparison, rather than a genuine algorithmic advantage.
> >
> > To resolve this, the updated version must explicitly clarify whether importance sampling was used for the DigiRL-STR baseline when sampling from the STR buffer. If it was not, you must either re-run the baseline with the proper correction or provide a thorough justification for its omission and transparently discuss its potential impact on the results.
> >
> > In summary, the paper's empirical results are strong enough. However, this updated score is conditional upon addressing the two points above to ensure the scientific integrity and clarity of the final publication.

---

> > > ### Author Response · Authors · 2025-08-07
> > >
> > > Thank you for taking the time to review our rebuttal and provide additional feedback. We appreciate your continued engagement with our work and address your remaining concerns below.
> > >
> > > 1. Training Stability
> > >
> > > We appreciate the reviewer’s concern regarding the distinction between anecdotal observations and substantiated evidence of training stability. While we did note in our rebuttal that SoLS appeared more stable than A2C during development, we agree that this point was not supported by sufficient empirical evidence and may have been overstated in our response.
> > >
> > > To address this concern, we will significantly tone down or remove any claims of enhanced training stability in the final version of the paper. This was not a central claim or contribution of our work, and we are happy to revise the language to reflect a more measured and accurate description of our findings. We will ensure the paper focuses on the core contributions and avoids any unsupported implications about comparative training robustness.
> > >
> > >
> > > 2. DigiRL-STR Baseline Implementation
> > >
> > > We thank the reviewer for this important methodological question and appreciate the opportunity to clarify our implementation rationale.
> > >
> > > We did not apply additional importance sampling corrections to DigiRL-STR when incorporating STR data. This was a deliberate choice, based on our understanding of DigiRL’s design and its fundamental differences from policy gradient methods like A2C:
> > > - **DigiRL is an off-policy algorithm** that operates on behavioural cloning principles with advantage-based filtering. It does not assume that training data is generated by the current policy, and in the original paper (which is for an off-policy method), no importance sampling correction is applied.
> > > - **A2C, by contrast, is an on-policy algorithm,** which assumes that the training data is drawn from the current policy. When augmenting A2C with off-policy data from STR, importance sampling is essential to correct for the distribution mismatch.
> > >
> > > Given this distinction, we did not apply importance sampling to DigiRL-STR, as doing so would introduce behaviour not present in the original algorithm and could alter its intent. In our view, applying importance sampling to DigiRL would not align with its original design and is not required for fair comparison, since DigiRL already operates in an off-policy regime.
> > >
> > > In contrast, A2C performs policy gradient updates that are theoretically grounded in on-policy assumptions. When using off-policy data, importance sampling corrections are necessary.
> > >
> > > That said, we fully agree that methodological clarity is essential. We will enhance Section 4.1.2 to explicitly explain this reasoning and add further clarifications. We appreciate the reviewer’s close attention to this issue and are committed to making this distinction more transparent in the final version. We also include further clarifications in the following part (3).
> > >
> > > 3. Additional Clarifications about DigiRL
> > >
> > > To further support the above, we would like to clarify our understanding of DigiRL and how that relates to our setup:
> > > - **Instruction value function and replay buffer:** The original paper uses an instruction-based value function to rank different episodes, with advantage serving as the metric for episode importance. A prioritised experience replay buffer stores previous trajectories. During training, trajectories are ranked based on the advantage from the instruction value function and sampled to create training batches.
> > > - **Step-level filtering:** During training, step-level advantages are computed (per equation (4.3) of the DigiRL paper), and only steps with advantage above a threshold (e.g., 0.05) are used for policy updates.
> > > - **Policy updates:** Steps that pass the filtering are used to update the policy by maximising the log-likelihood of the selected action.
> > >
> > > In our adaptation, we replace the original instruction value function and replay buffer with STR, which stores only successful steps, ensuring consistency across all baselines in our study. Note that the original DigiRL also does not apply importance sampling to replayed transitions.
> > >
> > > We acknowledge that our original phrasing may have caused confusion and will revise the text to reflect the implementation more precisely. We would also like to clarify that our implementation of DigiRL without STR should not be called ‘DigiRL’, and that it uses only fully on-policy trajectories. We will ensure that this point is clearly stated in the next version.
> > >
> > > We apologise for the lack of clarity and thank the reviewer once again for highlighting this issue.

---

### Decision · Program_Chairs · 2025-09-17

**Decision:**

Accept (poster)

**Comment:**

This paper introduces SoLS, a novel off-policy reinforcement learning algorithm for mobile app control. The authors combine an asymmetric policy update that favors positive outcomes with a Successful Transition Replay (STR) buffer that stores successful steps. Reviewers were largely positive, noting the method's strong empirical results and practicality, leading to a consensus for acceptance after the authors addressed concerns about novelty, stability, and experimental clarity in their rebuttal.